# A Patient with Synchronous Gallbladder and Bone Plasmacytoma

**DOI:** 10.3390/diagnostics13091509

**Published:** 2023-04-22

**Authors:** Mariam Markouli, Alexia Saridaki, Nora-Athina Viniou, Nefeli Giannakopoulou, Eleftheria Lakiotaki, Penelope Korkolopoulou, Panagiotis Diamantopoulos

**Affiliations:** 1First Department of Internal Medicine, Laikon General Hospital, National and Kapodistrian University of Athens, 11527 Athens, Greece; 2Department of Surgery, Euroclinic, 11521 Athens, Greece; 3Department of Pathology, National and Kapodistrian University of Athens, 11527 Athens, Greece

**Keywords:** extramedullary plasmablastic plasmacytoma, multiple myeloma, gallbladder mass, femoral bone, bone plasmacytoma

## Abstract

Multiple myeloma (MM) is the most common primary bone-originating tumor, whereas extramedullary plasmacytoma (EMP) is a plasma cell tumor that arises outside the bone and is most commonly found in the head and neck area. Gastrointestinal and particularly gallbladder involvement is exceedingly rare, and symptoms, if any are present, are usually similar to those seen with cholelithiasis. Treatment options usually include surgical resection and/or chemotherapy. In this report, we present a rare case of a clinically unexpected plasmablastic extramedullary plasmacytoma that was found on abdominal ultrasound (US) and magnetic resonance imaging (MRI) in a 61-year-old asymptomatic patient and led him to undergo cholecystectomy. A fluorodeoxyglucose positron emission computed tomography (FDG PET-CT) that was performed due to the onset of left thigh pain also demonstrated concurrent bone plasmacytoma. The patient is currently receiving chemotherapy and is also being prepared for autologous stem cell transplantation. In this context, we further present the diagnostic, therapeutic and prognostic challenges of EMPs. Lastly, we point out the distinct features of the plasmablastic subtype and analyze its differences compared to other histologic subtypes in achieving a successful diagnosis and management.

**Figure 1 diagnostics-13-01509-f001:**
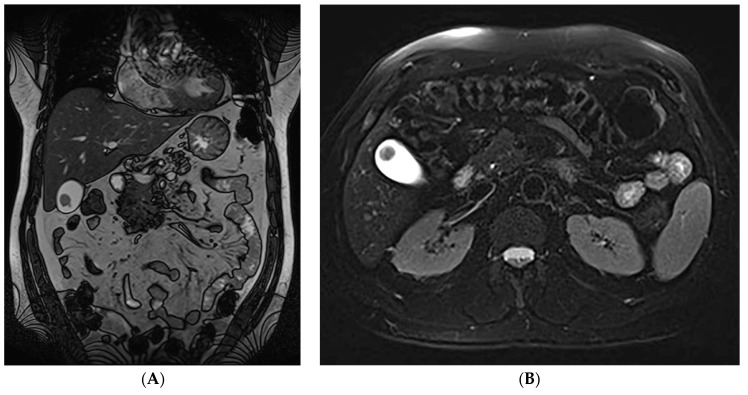
(**A**,**B**). Coronal (**A**) and transverse (**B**) T2- weighted images demonstrating a mass located near the gallbladder fundus. The accurate diagnosis of plasmablastic plasmacytoma is often difficult in routine clinical settings. Regarding cases of gastrointestinal involvement, the small intestine followed by the stomach are the two most common locations affected, whereas Extramedullary plasmacytomas (EMPs) of the gallbladder are exceedingly rare [1]. Multiple myeloma (MM) is the most common primary bone-originating tumor and is characterized by the malignant proliferation of a monoclonal plasma cell population, overproducing monoclonal paraproteins [2]. Extramedullary plasmacytoma (EMP) is a plasma cell tumor that arises outside of the bone and affects 4–6% of MM cases [1]. It can either exist in conjunction or independently of an underlying MM and is most commonly found in the head and neck area, such as the nasal cavity or paranasal sinuses, and less frequently in the gastrointestinal tract, pleura and lung, skin, lymph nodes, etc. [3]. In cases of concurrent multiple myeloma, the prognosis is poor, whereas isolated EMPs are usually low-grade tumors with relatively good prognosis, incidentally detected on imaging or postmortem [4]. Notably, the accurate diagnosis of plasmablastic plasmacytoma is often difficult in routine clinical settings due to its morphological and immunohistochemical overlap with plasmablastic lymphoma [5]. Regarding gastrointestinal involvement, the small intestine followed by the stomach are the two most common locations affected, whereas EMPs of the gallbladder are exceedingly rare [1]. Gallbladder EMPs may present with symptoms suggestive of cholelithiasis, acute cholecystitis and cholangitis, but may also be asymptomatic. Treatment options include surgical resection with adjuvant chemotherapy and/or radiation depending on their location, or autologous stem cell transplantation [1]. EMPs can present with cytomorphologic features ranging from mature to atypical, plasmablastic or anaplastic neoplastic cells. A plasmablastic morphology in particular has been associated with a higher proliferative index, a clinically more aggressive disease and poor survival [5]. A prompt accurate diagnosis, as well as the establishment of a more aggressive therapeutic approach, is therefore necessary for the management of patients with plasmablastic EMPs.

**Figure 2 diagnostics-13-01509-f002:**
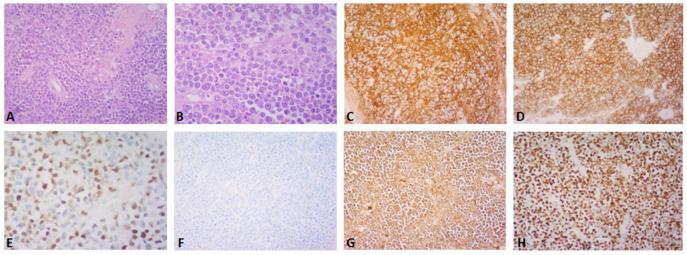
(**A**–**H**) Post-operative histopathology of the gallbladder lesion. (**A**) Hematoxylin-eosin stain, (×200): large cell population with diffuse growth pattern and complete effacement of the gallbladder wall (**B**) Hematoxylin-eosin stain, (×400): the neoplastic population shows plasmablastic morphology and abundant mitoses (**C**,**D**) CD138 and CD56 immunostain, (×200): the neoplastic cells are diffusely positive for CD138 and CD56 (**E**) cyclinD1 immunostain, (×400): nuclear positivity in ~50% of the cells (**F**,**G**) kappa and lambda light chain immunostain, (×200): the neoplastic cells are monotypic for lambda light chain (**H**) Ki67 immunostain, (×200): Ki67 proliferation index is ~95%. A 61-year-old asymptomatic male with a past medical history of benign prostate hyperplasia and gastroesophageal reflux disease was referred to our department for abdominal tenderness on deep palpation. An abdominal Ultrasound (US) demonstrated a clinically unexpected gallbladder mass, which was also confirmed on subsequent magnetic resonance imaging (MRI) (Figure 1A,B). The patient also underwent gastroscopy and colonoscopy at the time, which were unremarkable. On physical examination, mild right upper quadrant (RUQ) and epigastric tenderness were noted, the liver measured 2 cm on deep inspiration and the spleen was not palpable. Given the MRI finding, an elective cholecystectomy and a preoperative magnetic resonance cholangiopancreatography (MRCP) were scheduled, which confirmed the presence of a mass located near the gallbladder fundus. Post-operative histopathology led to the diagnosis of a plasmablastic plasmacytoma of the gallbladder (**A**–**H**). Around the time of his operation, the patient had also started experiencing left thigh pain. A fluorodeoxyglucose positron emission computed tomography (FDG PET-CT) that was performed one month post-surgery demonstrated a hypermetabolic left femoral bone lesion compatible with bone plasmacytoma (Figure 3A–D). Further bone marrow workup did not show evidence of underlying multiple myeloma.

**Figure 3 diagnostics-13-01509-f003:**
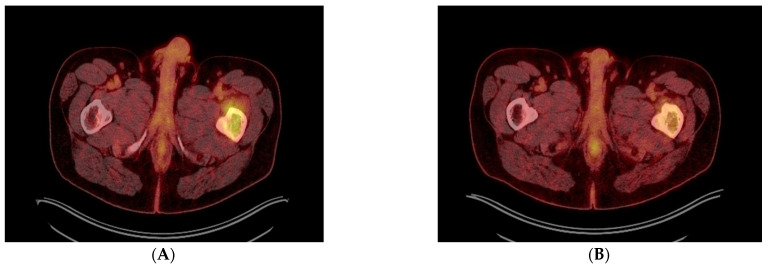
(**A**–**D**) Transverse (**A**,**B**) and coronal (**C**,**D**) PET-CT images before (**A**,**C**) and after (**B**,**D**) 4 cycles of DVRD, showing partial regression of the left femoral bone lesion. Besides undergoing a cholecystectomy, the patient was started on daratumumab, bortezomib, lenalidomide and dexamethasone (DVRd) and recently completed the fourth treatment cycle. A repeat PET-CT showed partial remission of the femoral lesion, and no signs of bone marrow (BM) involvement or recurrence in the gallbladder, as shown above. The patient is also currently being scheduled for an autologous stem cell transplant and continues to be followed up with the goal of completing nine chemotherapy cycles in total. In general, two thirds of patients with gallbladder pathology are found to have benign lesions, such as polyps [6]. Regarding malignant lesions of the gallbladder, more than 80% are reported to be adenocarcinomas, whereas only 28 cases of lymphoma have been reported in the literature, with the majority being Mucosa-associated lymphoid tissue (MALT) and diffuse large B-cell lymphomas [7,8]. Only 14 cases of gallbladder EMPs have been mentioned in the literature, with patients having a median age of 66 years and 50% of them having an underlying MM. Gallbladder EMPs may present with symptoms suggestive of cholelithiasis, acute cholecystitis and cholangitis, but may also be asymptomatic. Treatment options include surgical resection with adjuvant chemotherapy and/or radiation depending on their location or autologous stem cell transplantation [1]. Post-cholecystectomy histopathology of our patient’s lesion showed microscopic invasion of the gallbladder wall by a lambda-chain monoclonal population of CD138+, CD79+, CD56+, CyclinD1+ and IgA+ plasma cells with a significantly elevated Ki67 index of 98% and a diagnosis of plasmablastic plasmacytoma of the gallbladder was established. Of note, only one of 14 gallbladder EMP cases reported in the literature was shown to have plasmablastic features on histology with a Ki67 of more than 50%. Plasmablastic plasmacytoma is an undifferentiated round cell tumor that mainly contains plasmablasts and may resemble a plasmablastic lymphoma, sharing a nearly identical immunophenotypic profile with the latter [9]. Their main difference has been reported to be the fact that plasmablastic EMP is not associated with Epstein–Barr virus (EBV) [9,10]. It has been suggested that there could be a genetic overlap between these two tumors, since they are different manifestations of malignancies with a common B-cell derivation at an earlier different stage of maturation, a feature that conveys a plasmablastic morphology and thus a more aggressive clinical course [11]. This high grade of anaplastic cell morphology justifies a wide range of differential diagnoses, which include undifferentiated carcinoma, lymphoma, rhabdomyosarcoma, neuroendocrine tumor, neuroblastoma and melanoma. Our case is unique when compared to other reported gallbladder EMPs in that our patient was asymptomatic when diagnosed with plasmablastic EMP of the gallbladder with an extremely high Ki-67 index of 98%, compared to the only one plasmablastic gallbladder plasmacytoma reported in the literature. He was also subsequently diagnosed with synchronous plasmacytoma of the femoral bone without concurrent MM. When it comes to prognosis, the reported five-year overall survival rates of extramedullary EMPs are favorable, ranging from 78.4% to 87.4% [12]. Even though the exact prognostic percentages for gastrointestinal EMPs have not been determined, significantly lower survival rates have been reported for patients with intra-abdominal disease [12]. Despite the reported data that gastrointestinal involvement in EMPs equals a poor prognosis, especially when concurrent with MM, isolated implication of the gallbladder in most EMP cases appears to allow for good tumor remission when managed correctly. This is not the case, however, for plasmablastic EMPs, which are characterized by a high proliferation index and poor prognosis, as well as high recurrence rates. Treatment options of isolated gastrointestinal EMPs include surgical resection or cholecystectomy and potential stent placements through endoscopic retrograde cholangiopancreatography (ERCP) for symptom relief. Overall, it needs to be pointed out that in cases of underlying MM or plasmablastic EMPs, treatment needs to be more aggressive with supplemental immunochemotherapy and autologous stem-cell transplantation [13], even though concrete treatment recommendations have yet to be established.

## Data Availability

Not applicable.

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
