# Peer review of "A Patient with Synchronous Gallbladder and Bone Plasmacytoma"

_diagnostics, 2023, doi:10.3390/diagnostics13091509_

Round 1

Reviewer 1 Report (Previous Reviewer 2)

Corrections have been successfully made as suggested.

Reviewer 2 Report (Previous Reviewer 3)

The authors present the case of a 61-year-old male patient with synchronous gallbladder and bone plasmacytoma discovered incidentally during computed tomography. The patient was asymptomatic and underwent cholecystectomy after diagnosis.

The ''Discussion'' is deficient and is largely a repetition of the existing literature. This is not a discussion point, but should be concise, compare this case with similar case reports or case series in the literature, and highlight key differences from previously published studies. The fact that the patient was asymptomatic does not represent a significant difference from previously published reports.

Authors should prepare the literature review according to the Prisma 2020 guidelines. A literature review is an overview of the topic, an explanation of how publications differ from each other, and an examination of how each publication contributes to the discussion and understanding of the topic.

Finally, this is just a simple case with no new information, and similar pathology has been described several times in the literature. I see no benefit in this report for the readers of the journal. Even if the patient was asymptomatic at the time of diagnosis, the report of a simple case of extramedullary plasmacytoma is too premature to be published in an international journal.

Reviewer 3 Report (New Reviewer)

A manuscript by Markouli "A patient with synchronous gallbladder and bone plasmacytoma" presents interesting results. The manuscript is well and clearly written. The results are sufficiently discussed. The aim of the study is indicated. The references have been selected appropriately.

This manuscript is a resubmission of an earlier submission. The following is a list of the peer review reports and author responses from that submission.

Round 1

Reviewer 1 Report

The following article presents the atypical location of extramedullary plasmoblastic plasmocytoma in the gallbladder. It is unusual localization which can be surprising for intervening surgeon causing some anxiety.  The article is well written. The differential diagnosis is adequately supported by the literature.

Additional comments: The problem described in the paper is unique. Such an unusual localization of extramedullary plasmocytoma requiring surgery followed by immunochemotherapy is extremely rare. The knowledge of the possibility of unusual localization of the plasmocytoma could open alternative treatment options to avoid surgery. The case description is short but all the necessary clinical information were provided including detailed pathological analysis of the case. The authors discussed all possible differential diagnosis including of the plasmoblastic lymphoma. Detailed information of prognosis were provided. My impression is that the manuscript fulfils all requirements of the case report.

Author Response

We really appreciate the kind comments of Reviewer 1 and would like to thank them for spending their time to review our manuscript in detail. 

Reviewer 2 Report

In present study Markouli et al. described the rare case of Extramedullary plasmablastic plasmacytoma (EMP) of the gallbladder in a 61-year-old asymptomatic young patient.

Following are my suggestions to improve the article:

Major Remarks:

-          Given the uniqueness of the rare case of femoral bone plasmacytoma without concomitant multiple myeloma, it should be mentioned in the title as well otherwise it appears to be the usual EMP.

-          In the Conclusion section, please highlight future directions for research on this topic as evidence is now limited to case reports.

Minor Remarks:

-  Please remove the abbreviations that are not necessary and introduce the missing abbreviations (i.e., insert MM in the abstract line 11, the acronyms “MALT” line 74,”BM” line 60, “EBV” line 97, “ERCP” line 134, have never been mentioned and explained before).

- Keywords should be more specific, introducing “femoral bone” and “Extramedullary plasmablastic plasmacytoma”.

- The legends to the Figures are very confused, poor and not very descriptive. A greater expansion of the same is recommended to better understand the obtained results.

Author Response

We truly appreciate the valuable feedback of Reviewer 2 and thank them for their time and suggestions. All changes can be tracked in the revised document.

In addition, please find our point-by-point responses in red below:

Major Remarks:

-          Given the uniqueness of the rare case of femoral bone plasmacytoma without concomitant multiple myeloma, it should be mentioned in the title as well otherwise it appears to be the usual EMP.

We agree with Reviewer 2 and have changed the title accordingly.

-          In the Conclusion section, please highlight future directions for research on this topic as evidence is now limited to case reports.

After discussing with the Editor, the article type was changed from ‘Case Report’ to ‘Interesting Images’, according to the journal’s instructions, so there is no regular manuscript text (introduction/methods/results/discussion), just unstructured text followed by images with detailed legends. There is therefore no conclusions section, we have however included some future directions in the introductory text.

Minor Remarks:

-  Please remove the abbreviations that are not necessary and introduce the missing abbreviations (i.e., insert MM in the abstract line 11, the acronyms “MALT” line 74,”BM” line 60, “EBV” line 97, “ERCP” line 134, have never been mentioned and explained before).

Thank you for pointing this out. We have corrected the abovementioned abbreviations.

- Keywords should be more specific, introducing “femoral bone” and “Extramedullary plasmablastic plasmacytoma”.

Thank you, we have included more specific keywords.

- The legends to the Figures are very confused, poor and not very descriptive. A greater expansion of the same is recommended to better understand the obtained results.

As mentioned above, the article was changed to ‘Interesting Images’ and the layout was corrected accordingly. Please find the correct layout with the detailed and expanded figure legends in the revised document.

Reviewer 3 Report

The authors present the case of a 48-year-old man with a plasmablastic extramedullary plasmacytoma discovered incidentally during computed tomography. The patient was asymptomatic and underwent cholecystectomy after diagnosis.

The abstract is poorly organized; a more detailed account of the patient should be given. The patient's age should also be stated in the summary. The conclusions are general and do not arise from the case presented.

The introduction is very poor and not informative. The authors should give more details in the introduction about extramedullary forms of plasmacytomas, especially those in the GI tract. They should point out diagnostic dilemmas and include the differential diagnosis of this disease. Diagnostic and therapeutic modalities should be described in more detail.

The case presentation is not consistent with the data presented in the abstract. In the abstract, the author’s state that a plasmacytoma was found incidentally on computed tomography, whereas in a case description they state that a clinically unexpected gallbladder mass was found incidentally on magnetic resonance imaging.

The description of a case is also poor. Many important details are not presented, such as why the patient underwent MRI, what symptoms occurred, how long the symptoms lasted... Intraoperative findings were not even mentioned?!

Figure 2 was not even mentioned in the main text. Patohistological findings should be described in the case presentation (not in the discussion).

There is not even a word about the long-term follow-up of the patient.

Finally this is just a simple case, nothing new, and similar pathology has been described several times in the literature. I see no benefit to readers in this report. The report of a simple case of extramedullary plasmacytoma is too premature to be published in an international journal.

Search of literature is poor and should be improved.

Author Response

We would like to thank Reviewer 3 for their time and truly appreciate their valuable suggestions. Their comments were extremely insightful and helped us improve the quality of our manuscript.

All changes can be tracked in the revised document. In addition, please find our point-by-point responses in red below:

The abstract is poorly organized; a more detailed account of the patient should be given. The patient's age should also be stated in the summary. The conclusions are general and do not arise from the case presented.

We thank the reviewer for pointing this out, we have added a more detailed description of the patient, his age and more specific conclusions in the abstract.

The introduction is very poor and not informative. The authors should give more details in the introduction about extramedullary forms of plasmacytomas, especially those in the GI tract. They should point out diagnostic dilemmas and include the differential diagnosis of this disease. Diagnostic and therapeutic modalities should be described in more detail.

We agree and have included more details regarding extramedullary plasmacytomas, especially those arising from the GI tract. We have also expanded on the diagnostic and therapeutic challenges of the disease.

The case presentation is not consistent with the data presented in the abstract. In the abstract, the author’s state that a plasmacytoma was found incidentally on computed tomography, whereas in a case description they state that a clinically unexpected gallbladder mass was found incidentally on magnetic resonance imaging.

We apologize for this mishap and have corrected the reported data.

The description of a case is also poor. Many important details are not presented, such as why the patient underwent MRI, what symptoms occurred, how long the symptoms lasted... Intraoperative findings were not even mentioned?!

We agree and thank the reviewer for pointing this out. We have added a detailed explanation of the patient presentation and later findings.

Figure 2 was not even mentioned in the main text. Patohistological findings should be described in the case presentation (not in the discussion).

We have added a reference to figure 2 in the text. However, we describe the histopathology findings in the figure legend, as the article type was changed to ‘Interesting Images’ according to the Editor’s request and there is no included discussion based on the journal’s instructions.

There is not even a word about the long-term follow-up of the patient.

We thank the reviewer for pointing this out and have added more details on the patient’s long-term follow-up.

Finally this is just a simple case, nothing new, and similar pathology has been described several times in the literature. I see no benefit to readers in this report. The report of a simple case of extramedullary plasmacytoma is too premature to be published in an international journal.

Search of literature is poor and should be improved.

Given the article type that focuses on figures and not the report of a simple case or review of the literature, we are presenting interesting images of a very rare patient presentation, as no other case in the literature has reported a gallbladder EMP that is plasmablastic and has such a high Ki- index of ~98%. The rarity of this case also lies in the asymptomatic patient presentation, but also concurrency of bone plasmacytoma, as well as lack of concomitant multiple myeloma that successfully appeared to regress after chemotherapy.

Round 2

Reviewer 2 Report

Corrections have been successfully made as suggested.

Reviewer 3 Report

The authors revised the manuscript according to the reviewers' suggestions. Although this is a rare condition, it has been reported previously, so I do not see much benefit to readers in this report. It is at the discretion of the editor to decide whether this report merits publication in Diagnostics.
Also, the new title is totally inappropriate: Extramedullary gallbladder and synchronous bone plasmacytoma without concomitant multiple myeloma - what is a ''extramedullary gallbladder''??